# Lipocalin-2: A Nurturer of Tumor Progression and a Novel Candidate for Targeted Cancer Therapy

**DOI:** 10.3390/cancers15215159

**Published:** 2023-10-26

**Authors:** Maida Živalj, Jo A. Van Ginderachter, Benoit Stijlemans

**Affiliations:** 1Brussels Center for Immunology, Vrije Universiteit Brussel, 1050 Brussels, Belgium; 2Myeloid Cell Immunology Laboratory, VIB Center for Inflammation Research, 1050 Brussels, Belgium

**Keywords:** lipocalin-2, iron transport, tumor-associated macrophages, cancer, cell death

## Abstract

**Simple Summary:**

The aim of this review is to summarize the available information regarding Lipocalin-2 in tumor progression. Lipocalin-2 expression is correlated with tumorigenesis and thus considered as a biomarker in several subtypes of cancer. However, due to its key iron-regulating role, and the higher requirement of iron by cancer cells, it can also be considered as an additional target to improve the currently used cancer therapies.

**Abstract:**

Within the tumor microenvironment (TME) exists a complex signaling network between cancer cells and stromal cells, which determines the fate of tumor progression. Hence, interfering with this signaling network forms the basis for cancer therapy. Yet, many types of cancer, in particular, solid tumors, are refractory to the currently used treatments, so there is an urgent need for novel molecular targets that could improve current anti-cancer therapeutic strategies. Lipocalin-2 (Lcn-2), a secreted siderophore-binding glycoprotein that regulates iron homeostasis, is highly upregulated in various cancer types. Due to its pleiotropic role in the crosstalk between cancer cells and stromal cells, favoring tumor progression, it could be considered as a novel biomarker for prognostic and therapeutic purposes. However, the exact signaling route by which Lcn-2 promotes tumorigenesis remains unknown, and Lcn-2-targeting moieties are largely uninvestigated. This review will (i) provide an overview on the role of Lcn-2 in orchestrating the TME at the level of iron homeostasis, macrophage polarization, extracellular matrix remodeling, and cell migration and survival, and (ii) discuss the potential of Lcn-2 as a promising novel drug target that should be pursued in future translational research.

## 1. Introduction

In the past 20 years, cancer research has expanded and diversified enormously, whereby specifics and different manifestations of the disease have been investigated using high-throughput technologies, as well as computational and experimental tools, creating huge amounts of data, also referred to as “big data” on cancer research [1]. Based on these data, Hanahan and Weinberg, at the beginning of the 21st century, defined the hallmarks of cancer as a tool to identify the capabilities that normal human cells acquire on their way to becoming malignant cells [2]. The list of hallmarks of cancer expanded by an additional two by the year 2011, and a recent publication by Hanahan incorporated another four, bringing the total number to fourteen hallmarks of cancer [3,4].

Reprogramming or dysregulating cellular metabolism is one of the key elements of the hallmarks of cancer and is an essential aspect that tumor cells acquire to support accelerated cell proliferation and to allow for the maintenance of their enhanced biological functions. In this context, it was shown that glucose metabolism is one of the main dysregulated metabolic pathways that also leads to the dysregulation of protein and gene expression and, consequently, tumorigenesis, whereby cancer cells rely on aerobic glycolysis [3,5]. The uncontrolled proliferation of neoplastic cells is also dependent on the dysregulation of multiple other nutrients besides glucose, with iron being key to the maintenance of the cancer cells’ high metabolic demand [6,7,8]. Hence, alterations in iron metabolism are considered drivers of cancer cells’ aggressive behavior, including their uncontrolled proliferation, resistance to apoptosis, and enhanced metastatic ability. Moreover, dysregulated iron homeostasis has been associated with the development of an adverse tumor microenvironment (TME). In this context, macrophages, among other functions, play a very essential role in maintaining iron homeostasis, as they recycle, store, and release the iron into their surroundings. Within the TME, tumor-associated macrophages (TAMs), especially M2-like cells, contribute to cancer cell survival, promotion, and metastasis [9,10]. Previous studies showed that the iron-releasing phenotype is one of the defining characteristics of these M2-like macrophages, whereby Lipocalin-2 (Lcn-2) serves as a carrier protein [11,12]. Lcn-2, also known as neutrophil gelatinase-associated lipocalin (NGAL), oncogene 24p3, siderocalin, or uterocalin, is a 25 kDa glycoprotein that was initially discovered as part of the innate immune system, but was later discovered to deliver iron to the fast-growing tumor cells within the TME [13,14]. Macrophage polarization is closely associated with the differential regulation of iron metabolism, whereby an increase in TAM’s Lcn-2 expression causes an iron release phenotype that supports tumor growth and therapy resistance [15,16,17].

Another hallmark of cancer in which Lcn-2 could play a role focuses on the apoptotic resistance of cancer cells. Cancer cells are capable of circumventing apoptosis by, for example, losing the function of tumor suppressor genes such as tumor protein 53 (TP53), by downregulating proapoptotic factors such as Bim, Bax, and Puma, by increasing the expression of survival signals such as insulin-like growth factor 1 and 2 (Igf1/2), or by upregulating antiapoptotic proteins such as Bcl-x_L_ and Bcl-2 [3]. Moreover, the deprivation of trophic factors, such as IL-3, can also lead to apoptosis, whereby Lcn-2, being transcriptionally regulated by IL-3 deprivation, induces apoptosis [18,19]. Additionally, Lcn-2, being an alternative regulator of iron homeostasis, was also shown to play a role in an iron-dependent, non-apoptotic mode of cell death called ferroptosis, which is characterized by alterations in intracellular iron levels and lipid peroxidation [20]. Important to mention is that ferroptosis is driven by disordered iron metabolism, resulting in the production of reactive oxygen species that induce the Fenton reaction and/or impair mitochondrial iron metabolism [21]. The upregulation of iron transport due to inflammation can cause lipid accumulation and oxidative stress, which is further fueled by Lcn-2′s ability to internalize iron [22].

High expression of Lcn-2 in cancerous tissues of the thyroid, ovarian, breast, prostate, pancreatic, renal, and colorectal organs underline the tumor-promoting role of Lcn-2. This is attributed to the fact that Lcn-2 is also able to promote epithelial-to-mesenchymal transition (EMT) and bind to matrix metalloprotease 9 (MMP9), thereby modulating the metastatic phenotype of cancer cells. In addition, Lcn-2 was shown to contribute to the polarization of macrophages and, in turn, to promote iron delivery to cancer cells, whereby the increase in intracellular iron protects cancer cells from apoptosis [23,24,25,26]. Overall, this review will further discuss the implication of Lcn-2 in cancer development, especially in breast carcinoma, focusing on the role of Lcn-2 in the context of TAM polarization via iron regulation, as well as on the effect of Lcn-2 on cell death and the potential of utilizing this concept to improve immunotherapy.

## 2. Role of Iron in Cancer Progression

Iron is involved in many important functions in mammalian cells, such as cell metabolism, proliferation, and growth [6]. More than half of the total amount of iron is stored in erythrocytes as part of hemoglobin, providing oxygen transport throughout the body. Besides this, it is also used as a helper molecule for an array of proteins, playing a key role in cell cycle progression, DNA synthesis and stability, oxidative phosphorylation, and the citric acid cycle [6]. Due to the low availability of iron in circulation, proteins involved in the import, storage, and export of iron need to be highly regulated. Although iron plays an essential role as a cofactor for enzymes, due to its ability to lose and gain electrons, it can also be potentially dangerous as it can play a role in free-radical reactions, which can be mutagenic [27,28]. One of these reactions is the Fenton reaction, whereby ferrous iron (Fe^2+^) donates an electron to hydrogen peroxide, making a hydroxyl radical, which is an oxidant of reactive oxygen species (ROS). ROS can induce lipid and protein modifications in cancer cells, leading to aggressiveness and metastasis. Conversely, the accumulation of iron-dependent lipid modifications can lead to ferroptosis (i.e., iron-dependent cell death), which will have a tumor-suppressing role [29,30]. Hence, iron-regulation plays a pivotal role in cancer by exerting both tumor-promoting functions and tumor-suppressing functions. In addition, a lack of iron can cause anemia, which, in turn, has a tumor-promoting effect by causing hypoxia that fuels M2 polarization [31]. In this context, the occurrence of cancer-related anemia (CRA) is one of the most frequent secondary problems and is linked to disease progression (i.e., the occurrence of metastasis) as well as the tumor site and age of the patient [32]. In fact, CRA is a cytokine-mediated disorder resulting from complex interactions between tumor cells and the immune system and is characterized by biological and hematologic features that resemble those described in anemia associated with chronic inflammatory disease (i.e., anemia of inflammation) [33]. It was also shown that the release of certain inflammatory cytokines during anemia of inflammation negatively affects erythroid progenitor cell differentiation. In this context, tumor cells can produce cytokines that negatively affect erythroid progenitor cell differentiation. Moreover, Lcn-2 has been shown to also affect erythropoiesis negatively, and this is related to the occurrence of hypoxic TAMs and the development of metastasis [34,35,36]. Hence, Lcn-2, through its iron-regulating role, could be implicated as a pivotal player in CRA by regulating the inflammatory immune response within the TME and iron availability for erythropoiesis [37,38,39]. While carcinogenesis is known to cause a decrease in the production of red blood cells, the main cause of anemia during cancer is a consequence of radio- or chemotherapy, which are immunosuppressive and often damage erythroid progenitors and/or reduce erythrocyte half-life [40].

Dysregulated iron homeostasis is associated with the malignant cancer phenotype, whereby the change in the expression of iron-regulating genes promotes higher metabolic needs of cancer cells [41]. The key players in iron homeostasis are (i) transferrin (Tf), which is the main iron transporter that, upon binding to the transferrin receptor 1 (TfR1), allows for iron import; (ii) heme oxygenase 1 (Hmox1), which plays a key role in iron regulation and ROS production; (iii) divalent metal transporter 1 (DMT1), which allows for intracellular iron transport from the endosome to the cytosol; (iv) ferritin (FT), which is essential for iron storage; and (v) ferroportin (FPN), which is the sole iron exporter regulated by hepcidin [42,43,44,45]. TfR1, FT, DMT1, Hmox1, and hepcidin levels are highly upregulated in breast, prostate, lung, and squamous cell carcinoma, while FPN levels are significantly lowered compared to those in healthy tissue [46,47,48]. Simultaneously, iron is highly internalized and stored by cancer cells, while its efflux is halted by the dysregulation of FPN expression [48]. Furthermore, changes in iron levels also regulate a multitude of tumor-suppressive signaling pathways, such as those related to tumor protein 53 (p53), c-myc, nuclear factor erythroid 2-related factor 2 (NRF2), Harvey rat sarcoma virus (H-RAS), signal transducer and activator of transcription (STAT3), extracellular signal-regulated kinase ½ (ERK1/2), protein kinase B (AKT), and hypoxia-inducible factors 1α and 2α (HIF1α and HIF2α) [7]. Overall, it is clear that cancer cells demand a higher uptake of iron, which cannot be mediated solely via the Tf-TfR-mediated pathway. Therefore, Lcn-2, a transferrin-independent iron carrier, allows cancer cells to acquire the additional necessary iron. This process of non-transferrin-mediated iron uptake, involving Lcn-2 as a key player, has been shown to be involved in cancer progression [49]. However, Lcn-2 is unable to bind to iron directly, but it can bind to the iron–siderophore, or siderophore-like, complex, and can be internalized by the high-affinity cell surface receptor SLC22A17 (24p3R) [50].

Bauer et al. assessed Lcn-2 expression in a representative cohort of 207 breast cancer patients, whereby a strong association was found between Lcn-2 expression and prognostic factors such as the Ki-67 proliferation index, lymph node involvement, and human epidermal growth factor receptor 2 (HER-2/neu) status and histological grade [51]. Although there is no correlation between Lcn-2 and spontaneous polyoma-middle-T oncogene (PyMT) breast cancer parameters, Mertens and colleagues reported a positive relationship between the expression of Lcn-2 and tumor onset, lung metastases, and recurrence. This was mediated by stromal cells, mainly macrophages, whereby Lcn-2^−/−^ TAMs stored more iron compared to wild-type TAMs [17,52]. A study by Tymoszuk and colleagues showed a negative influence on the efficacy of different immunotherapies after iron administration. Indeed, supplementing E0771 triple-negative breast cancer-bearing mice with isomaltoside, an iron compound typically used for the treatment of patients with iron deficiency (linked to anemia), promoted tumor growth by negatively impacting T-cell-mediated immune function and infiltration, also impairing the efficacy of anti-PD-L1 and IL-2/doxorubicin immunotherapies [53]. Conversely, iron-chelating therapies (e.g., deferoxamine, DFO) were shown to decrease cancer cell growth in a leptomeningeal metastasis model. It was shown that macrophages provide signals that trigger Lcn-2 production by cancer cells, which, in turn, allows cancer cells to outcompete macrophages in acquiring iron by using Lcn-2 as alternative iron supply. Yet, DFO treatment was shown to impair the tumor growth and shRNA of Lcn-2, and its receptor was shown to impair the iron acquisition and proliferation of tumor cells [54]. A relationship between clear-cell renal cell carcinoma (ccRCC) and Lcn-2 in the context of iron regulation was studied by Rehwald et al., which provided new insights into the contribution of iron-loaded Lcn-2 in matrix adhesion and migration. It was shown that patients suffering from ccRCC have an elevated level of the iron-loaded form of Lcn-2. By stimulating human patient-derived tumor cells (T-TEC) in vitro with a mutant, non-iron-binding Lcn-2, the iron-binding capacity was disabled in comparison to normal non-modified protein, resulting in a significant reduction in the intracellular iron amount. Importantly, adhesion to collagen I or fibronectin matrices, which are crucial for cancer cell migration and matrix adhesion, were inhibited using non-iron-binding Lcn-2 [55].

Finally, previous years focused on targeting the iron regulation mechanism as a potential anticancer therapy. This is true for both iron chelator-based therapies, such as the already mentioned DFO, deferasirox (DFX), ciclopiroxolamine (CPX), Vlx600, Dpc, and thiosemicarbazone, as well as for iron trafficking-based therapies such as the targeting of hepcidin, FPN, and TfR1 [42,56]. The former strategies lead to a lot of side effects due to poor cancer cell targeting in the TME, whereby a lot of nonmalignant cells are affected, leading to strong cytotoxicity. The latter faced ambiguous results. Some promising results were obtained by targeting the hepcidin/IL-6 axis, or the hepcidin/IL-8 axis, leading to decreased metastasis of breast cancer to the liver, lymph nodes, and lungs, while anti-TfR antibodies were only effective in some patients and in certain cancer subtypes, whereby anemia was reported as an important side-effect [57,58,59]. However, recent improvements and optimization for targeting the Tf-TfR system are still considered promising, especially in multidrug-resistant tumor cells [60]. Nevertheless, as previously indicated, cancer cells can go beyond the classical iron uptake pathway to allow for iron scavenging from the surroundings. Hereto, the Lcn-2/Lcn-2R pathway could be an interesting target, whereby blocking of the iron-binding region or receptor-binding region of Lcn-2 would lead to reduced iron availability in the TME.

### 2.1. Lipocalin-2 as an Alternative Iron Regulator

Lipocalin-2 belongs to the lipocalin superfamily, whose members are involved in a very diverse set of functions, such as the regulation of cell division, metabolic homeostasis, cell-to-cell adhesion, and differentiation, but they predominantly function as carriers or transporting vehicles of small/low-molecular-weight lipophilic molecules. Though their amino acid sequence identity is quite diverse, they all share a common secondary and tertiary structural feature called “the lipocalin fold”. The lipocalin fold consists of eight antiparallel β sheets, bonded throughout the fold with hydrogen bonds to one another, thereby making a β-barrel. Seven short hairpin loops connect the β sheets to one another, whereby one of the loops creates a lid-like structure that can close the ligand-binding cavity. The other end of the β-barrel, i.e., the N-terminal part, is closed by a short 3_10_-helix. The cavity made from this structure is a cup-shaped calyx where the internal binding of specific ligands is possible. The region closest to the ligand binding site, which contributes to its activity, is composed mainly of hydrophobic residues, while the region closest to the opening of the barrel is made up mostly of polar residues. Lcn-2′s cavity, broader and shallower in comparison with other proteins from the same family, can bind to mammalian and bacterial proteins called siderophores, which are able to form a complex with circulating iron [61].

Lcn-2 can be found in three different molecular forms, namely, a 25 kDa monomer, a 45 kDa homodimer, and a 135 kDa heterodimer with MMP9 (92 kDa) [62]. The monomeric form of Lcn-2 is shown to be a potent iron chelator, while the dimeric variant is unable to chelate iron, yet has a longer half-life and is mainly released by neutrophils and increased upon inflammation [63]. The heterodimer Lcn-2/MMP9 was shown to be elevated in patients suffering from different carcinomas [64]. However, in mice, Lcn-2 cannot form a heterodimer with MMP9 due to the lack of a Cys87 residue that allows for disulfide bridge formation between Lcn-2 and MMP9, which makes it impossible to study this complex in murine models [65]. So far, six putative Lcn-2 receptors have been identified, exhibiting different affinities for Lcn-2, namely, neutrophil gelatinase-associated lipocalin receptor (NGALR, 24p3R), low-density lipoprotein-related protein 2 (LRP2), LRP6, melanocortin 4 receptor (MC4R), MC1R, and MC3R [66]. Important to mention is that most of these receptors also bind other ligands, which complicates the identification of the precise receptor activities and responsible pathways that are triggered following Lcn-2 binding and internalization. Among these receptors, the best studied are megalin, also known as LRP2, which exhibits a low-affinity for Lcn-2 (~60 nM), and solute carrier SLC22A17, also known as 24p3R, exhibiting a high affinity for Lcn-2 (~92 pM) [19,67]. Yet, depending on the iron-binding status, the affinity of Lcn-2 for 24p3R varies from 7–10 µM (apoLcn-2 to NGALR aa1- aa105) to ~20 µM (Lcn-2/ferric enterobactin to NGALRaa1- aa105) [68]. Megalin is mainly expressed by kidney epithelia, whereas 24p3R is expressed in different tissues. The binding of Lcn-2 to 24p3R triggers different transcription factors, such as nuclear factor kappa-light-chain-enhancer of activated B cells (NF-κB), activator protein 1 (AP-1), GATA-1, PU.1, and CCAAT/enhancer-binding protein β (C/EBPβ) [50,69]. As previously mentioned, upon bacterial infection and recognition by innate immune cells, Lcn-2 is secreted and can bind to bacterial iron-loaded siderophores. Subsequently, this complex is transported through the Lcn-2 receptor (24p3R) into mammalian cells where iron is stored. Hence, Lcn-2 has a bacteriostatic effect by limiting bacterial iron availability. In addition, Bao and colleagues discovered that siderophore-like proteins also exist in mammals and that these commonly occurring metabolites called catechols can also bind to Lcn-2 inside the mammalian body. However, it was shown that enterobactin (i.e., a high-affinity iron binder) inhibits the binding of catechol to Lcn-2, suggesting that bacterial siderophores compete with endogenous mammalian siderophore-like proteins to bind free iron [13]. The first mammalian siderophore that was identified is called 2,5-dihydroxylbenzoic acid (2,5-DHBA), which binds specifically ferric iron (Fe^3+^). Bacterial enterobactin and 2,5-DHBA are chemically similar and differ by only one hydroxyl group; hence, it is expected for both to bind to the shallow and broad Lcn-2 calyx [70]. Since pathogens take up siderophore-bound iron, mammals secrete Lcn-2 as a defense system, as it will form a tight complex with the iron-loaded catechol and hamper the uptake by pathogens.

Interestingly, Lcn-2 is upregulated in various tissues and fluids in the body, and is linked to several diseases, including ischemic, inflammatory, and metabolic disorders, as well as cancer. Lcn-2 was also shown to be elevated in cases of cerebrovascular accidents, such as stroke and myocardial infarction [71]. Furthermore, Lcn-2 is considered a real-time, sensitive biomarker for renal diseases and not only plays a role as a biomarker for disease states, but in certain cases, also actively protects against acute kidney injury (AKI), such as ischemia–reperfusion injury [72]. Lcn-2 is also considered, from a metabolic standpoint, to be an adipokine, since it is able to promote insulin resistance due to its high expression in adipose tissue. It was elevated in serum and in adipose tissues in disorders connected to obesity, in both murine models and in patients [73]. Therefore, the pathways triggered by Lcn-2, besides the iron transporting pathway, depend on the disease context. In addition, for some diseases, such as stroke–reperfusion injury, monoclonal anti-Lcn-2 antibodies are used as a potential therapeutic modality [74].

As the expression of Lcn-2 is mainly altered during different inflammatory or malignant conditions, it is suggested that the upregulation of Lcn-2 only occurs under pathophysiological conditions, whereby iron homeostasis is compromised as an efficient defense mechanism to impair iron uptake by pathogens. However, it is important to note that the main iron uptake pathway mediated by transferrin via the transferrin receptor is predominant during physiological conditions. Yet, during different inflammatory conditions or malignancies such as cancer, the altered expression of the TfR1 has also been linked to disease progression; hence, this pathway can also be considered a potential therapeutic target [75,76]. Within this review, we will mainly focus on Lcn-2 and its role in iron regulation. More specifically, in the following sections, we will focus on the macrophage-mediated regulation of Lcn-2 expression and the effect of Lcn-2 on macrophage polarization in cancer development and metastasis.

### 2.2. Polarization of Tumor-Associated Macrophages by Iron

Macrophages (MΦ) are specialized innate immune cells that play an important role in host defense by quickly responding to different danger signals through the initiation of inflammation, the recognition of pathogens, antigen presentation, as well as the regulation of homeostasis at different levels. Among the various homeostatic functions of macrophages, they are found to play a central role in iron homeostasis during both steady state and disease, with an essential role in iron recycling [77,78].

Upon inflammation or tissue injury, monocytes rapidly extravasate from the bone marrow into the blood and subsequently migrate into the target tissue, where they will differentiate into mature macrophages or dendritic cells depending on the environmental signals they receive. For macrophages, these signals will be essential to promoting their polarization and allow these cells to exert different functions. Some of these functionalities are migration along a chemokine gradient, the clearance of apoptotic cells, the production of immune-regulatory molecules, antigen presentation, participation in the onset and resolution of inflammation, and involvement in innate and adaptive immunity [79]. This way, macrophages can optimize their performance and adapt to different tissue demands.

Classically activated macrophages (M1), also called pro-inflammatory macrophages, are induced by Th1 (T cell helper cell) cytokine interferon-γ (IFN-γ), Toll-like receptor (TLRs) agonists, and bacterial moieties such as lipopolysaccharide (LPS). Upon activation, M1 cells are characterized by their induction of major histocompatibility complex (MHC) class II, but also class I, the formation of nitrogen species (NO) and reactive oxygen (ROS), and the expression of pro-inflammatory cytokines such as tumor necrosis factor α (TNF-α) and interleukins (i.e., IL-6, IL-23, IL-1β, and IL-12). Alternatively activated macrophages (M2), also called anti-inflammatory macrophages, act in an opposing way. The activation of such macrophages is driven by IL-4, IL-13, and T helper 2 (Th2) cells, whereby their characteristic cytokines and features are a high expression of IL-10, arginase-1, mannose receptor-1, transforming growth factor (TGF) β, and prostaglandin E2 production [10,80].

In the context of cancer, the polarization status of macrophages has been shown to play a key role in the progression of the disease. Indeed, these cells’ role in regulating iron-homeostasis was shown to be very important during cancer progression (see above), whereby M2-like macrophages are known for their iron release phenotype. To do so, they upregulate the expression of TfR, CD91, CD163, Hmox1, FPN, DMT1, and iron-regulated RNA binding proteins (IRP), while downregulating FT expression, which allows for the provision of iron to the tumor cells. Moreover, the iron-donating phenotype of M2-like macrophages and their role in tumor promotion are associated with poor patient prognosis, tumor size, and aggressiveness [81].

As mentioned before, Lcn-2-mediated iron transport is an alternative way to acquire iron, especially during inflammation. Since macrophages are key players in iron homeostasis, iron sequestration (which is reminiscent of M1-like macrophages) will be the first response to foreign cell recognition (since tumor cells are initially recognized as foreign) and is thus required to prevent tumor growth. This will be followed by competition between immune and cancer cells for this essential nutrient. To survive, cancer cells release factors that affect the polarization of TAMs into a phenotype that favors iron release. Thus, cancer cells instruct TAMs to adopt an M2-like anti-inflammatory phenotype (Figure 1) [42,82]. Interestingly, within the TME, heterogenous populations of macrophages (ranging from M1 to M2) can be found, both retaining specific phenotypes, and both are regulated under local stimuli [83]. Recent studies using single-cell (sc) RNA sequencing technology unraveled higher diversity in macrophage subsets. Thereby, not only can M1 macrophages that are characterized by their MHCII^high^ expression and M2-like macrophages that are characterized by their MHCII^low^ expression co-exist within the TME, but also, new subpopulations emerged from these sc-data. For instance, LYVE^+^, CD169^+^, and FOLR2^+^ macrophages do not fit within either M1 or M2 signatures [84,85,86,87]. Accordingly, it can be added that TAM polarization is an ongoing process that is affected by functional activation phenotypes [88,89,90]. Overall, a vast array of preclinical studies have identified numerous pathways that are critical in the polarization, recruitment, and metabolism regulation of TAMs during all stages of tumor progression [91,92]. These studies highlight the importance of targeting the immunosuppressive phenotype of TAMs and might offer promise for improved cancer therapy.

Due to the higher demand of iron, it is suggested that tumor cells hijack macrophages to turn them into an iron delivery source. Studies have shown that Lcn-2 plays a very important part in the pro-tumorigenic macrophage phenotype. Indeed, the uptake of dead tumor cells, which died due to apoptotic, accidental necrotic and necroptotic cell death induced by professional phagocytes, results in a functional shift towards an alternative (M2-like) phenotype, coinciding with the stimulation of the expression and secretion of Lcn-2 (Figure 1) [93]. Thus, the uptake of dying cells also creates a macrophage phenotype in favor of iron release [94]. Mechanistically, sphingosine-1-phosphate (S1P) released from apoptotic cancer cells will, upon signaling through its receptor, S1PR1, enhance Lcn-2 production by TAMs, which, in turn, release the lymphangiogenic factor VEGF-C to promote tumor growth [69,95].

Agoro and colleagues have shown that the increased loading of iron in macrophages promotes the expression of polarization markers reminiscent of an M2-like phenotype, while the opposite observation was made when macrophages faced iron deficiency [96]. Indeed, the increase in tissue iron deposition and TF saturation was caused by an increased iron status, resulting in the stimulation of M2-associated cytokines and markers such as *Ym1* and *arginase*, while the iron chelator deferoxamine decreased MHCII expression. Marques and colleagues reported high levels of iron in macrophages, with TfR and FPN upregulation in breast cancer tissues, while Leftin et al. reported that the depletion of iron-laden TAMs, using small-molecule inhibitors of the macrophage colony stimulating factor 1 receptor (CSF1R) or anti-CSF1R IgG in a murine breast tumor model, slowed mammary tumor growth, leading to the conclusion that iron deposition in macrophages can contribute to their tumor-supportive role [97,98]. Further, lymphangiogenesis and angiogenesis are promoted by M2-like (high ratio of CD163^+^/CD68^+^) TAMs, whereby a high expression of VEGF-C and VEGF-A by M2-like macrophages was reported in patients with non-small-cell lung cancer (NSCLC) [99].

Overall, the increased turnover of iron by M2-like TAMs is beneficial not only for cancer cells, but also for tumorigenesis in general. Thereby, the growth characteristics of neighboring cells, such as fibroblasts and other tumor cells, in the TME are upregulated by the increased iron availability. Therefore, an intriguing hypothesis could be that iron from the increased labile iron pool in M2-like TAMs is released into the TME via, among other factors, Lcn-2. Although during the onset of carcinogenesis, macrophages behave as wound healers and repairers of the tissue, soon, this growth-supportive phenotype, alongside the increased Lcn-2 expression, is controlled by cancer cells, which will ultimately benefit from the surplus iron donated by macrophages via Lcn-2.

## 3. Lipocalin-2 in Cancer Progression

The involvement of Lcn-2 in carcinogenesis has been studied using murine models, in human and murine cancer cell lines, and in patients. One of the malignancies in which Lcn-2 has been most studied is breast cancer, where the increased expression of Lcn-2 in carcinoma tissue, urine, and sera correlates with a poor prognosis and increased aggressiveness. The study conducted by Provatopoulou revealed that Lcn-2 plays a heterologous role in the development of breast carcinoma. It was shown that Lcn-2 serum levels did not differ in women with benign breast conditions that might lead to breast cancer, such as atypical ductal hyperplasia, ductal carcinoma, and sclerosing adenosis, compared to healthy controls. However, there was a significant increase in Lcn-2 expression in patients with invasive ductal carcinoma (IDC), which is the most common type of invasive breast cancer, whereby a significant positive association was found between the disease severity score and Lcn-2 expression in serum [100]. Compared to other breast carcinomas, triple-negative breast cancer (TNBC) has more aggressive tumor progression and worse prognosis, whereby the metastasis of this cancer subtype leads to a 5-year survival rate of only 10.8% [101]. It was found that Lcn-2 expression can be considered an independent prognostic biomarker for the reduced survival of breast cancer patients, particularly those suffering from TNBC. Of note, for the more rare but most aggressive and deadly variant of primary breast cancer, i.e., inflammatory breast cancer (IBC), high levels of Lcn-2 have also been associated with a poor prognosis and reduced overall survival. A relationship between other markers of poor breast carcinoma prognosis, such as progesterone receptor (PR)- and estrogen receptor (ER)-negative status and Lcn-2, has been reported in primary breast carcinoma. However, heterogenous expression of Lcn-2 at protein and mRNA levels was also described by Stoesz, whereby Lcn-2 was detected in 42.2% of patients [102,103]. Furthermore, patients in stages II and III were reported to have increased expression of Lcn-2 in the tumor stroma, compared to healthy tissue, and patients with metastatic breast cancer were reported to have increased expression of Lcn-2 in the urine [104]. Guo and colleagues showed that the siRNA silencing of Lcn-2 in a TNBC model inhibited angiogenesis in vivo and in vitro, while in a similar study on IBC, the depletion of Lcn-2 in cell cultures reduced invasion, migration, and the cancer stem cell population [105,106]. Furthermore, secreted factors by four stromal components (fibroblasts, lymphatic endothelial cells, macrophages, and blood microvascular endothelial cells) were screened upon stimulation with conditioned medium from four different TNBC cell lines (MDA-MB-231, SUM159, MDA-MB-468, and SUM149). The results showed that Lcn-2, together with IL-6, CCL5, and IL-8, was significantly upregulated in the crosstalk between four different TNBC cell lines and stromal cells [107]. Finally, core biopsies of 652 breast cancer patients undergoing neoadjuvant chemotherapy, examined via immunohistochemistry, revealed that the intensity and expression of Lcn-2 were significantly related to estrogen and progesterone receptor status, as well as with the histological tumor type [103].

Lcn-2 was also found to be upregulated in residual cancer cells, found in the host after chemotherapeutic treatment that causes senescence of cancer cells. These senescent cells release a set of pro-inflammatory chemokines, cytokines, and growth factors, which collectively are referred to as the senescence-associated secretory phenotype (SASP). Additionally, the inactivation of Lcn-2 by CRISPR/Cas9 gene deletion increases the response to chemotherapy in murine breast cancer. Importantly, it was shown that neoadjuvant therapy leads to the upregulation of Lcn-2 in human breast tumors, highlighting the importance of targeting Lcn-2 as an additional therapeutic approach [108]. Studies on murine and human cancer cells revealed a correlation between breast carcinoma progression and Lcn-2 expression. For instance, using the well-established polyomavirus middle T antigen (MMTV-PyMT) breast carcinoma model, it was shown that crossing MMTV-PyMT mice with Lcn-2^−/−^ mice resulted in a decreased tumor onset and burden compared to wild-type MMTV-PyMT mice. Interestingly, discoveries using experimental mouse and human cell lines correlate with findings in patients [109,110,111].

Also, in other cancer types, Lcn-2 was reported to promote tumor progression. The expression level of Lcn-2 in the bile of cholangiocarcinoma (CCA) patients was significantly higher than in control groups, while Lcn-2 knock-down inhibited cell growth in vitro and in vivo, while the overexpression of Lcn-2 increased the cell metastatic potential, making it overall a potentially prognostic marker for this disease [112]. Lcn-2 was also identified as an important suppressor of radiotherapy success in oral squamous cell carcinoma (OSCC), whereby radiated Ca9-22 cells showed the strongest increase in Lcn-2 expression. The radiosensitivity was also increased in lung carcinoma upon Lcn-2 knock-down by siRNA [16]. A mouse model of hepatoblastoma (HB) was reported to have high expression levels of Lcn-2, with the highest rate in its embryonal form. In most human HB samples, Lcn-2 is highly expressed, and there is a correlation between Lcn-2’s presence and the histological subtype within individual tumors. Whilst there are some conflicting data on the role of Lcn-2 and its inhibition of epithelial-to-mesenchymal transition (EMT) in hepatocellular carcinoma, Molina et al. suggested that hepatocyte-derived Lcn-2 could serve as a potential serum biomarker in HB [113].

Overall, numerous studies have shown that Lcn-2 facilitates tumorigenesis by enhancing tumor cell growth and survival, and by increasing cellular resistance to chemotherapeutics and iron-induced toxicity [16,114]. Mechanistically, the oncogenic role of Lcn-2 is associated with its ability to make a complex with MMP9. The overexpression of Lcn-2 and MMP9 is associated with an early promoter methylation status, leading to the development of primary tumors [115]. Other studies proposed that the contribution to tumor metastasis is attributed to Lcn-2′s ability to promote EMT, which is a central process in cancer cell dissemination [24,116].

However, recent studies showed that Lcn-2 can have both a pro- and anti-tumorigenic role depending on the tumor stage, type, and location. This opposing role of Lcn-2 appears to be regulated in an iron-dependent manner, whereby holo-Lcn-2 (i.e., iron–siderophore-loaded Lcn-2), can fuel tumor growth, while apo-Lcn-2 (i.e., iron–siderophore-complex-free Lcn-2), promotes apoptosis. Tong et al. showed that Lcn-2 overexpression significantly blocked pancreatic cancer cell invasion and adhesion and potently decreased angiogenesis in vitro, yet it did not affect cancer cell viability and survival [117]. Also, Lcn-2 expression substantially inhibited liver metastasis upon inoculation of nude BALB/c mice with a human highly metastatic liver cancer cell line, KM12SM, thereby proving that Lcn-2 can also have a negative effect on tumor development [118]. However, the literature on the tumor-suppressive function of Lcn-2 is limited, and less frequently reported [65].

## 4. Cell Death and Lipocalin-2

One of the mechanisms contributing to drug resistance and treatment failure in cancer cells is their ability to alter cell death pathways. The homeostasis of an organism is, among various mechanisms, dependent on the dynamic production and elimination of cells. Though many forms of cell death exist, the best characterized form is apoptosis, which was shown to mediate tumor regression following chemo/radiotherapy [119]. Uncontrolled tumor cell initiation and proliferation, which are some of the main phenotypes of malignant cells, could be led by the inactivation of pro-apoptotic proteins or the expression of anti-apoptotic factors, which was reported by different groups [120]. On the other hand, the modulation of ferroptosis, a form of cell death related to iron availability in the cell, also resulted in inhibition of the migration and proliferation of cells [121,122]. Intriguingly, Lcn-2 was reported to be involved in both apoptotic and ferroptotic mechanisms, whereby a central player is iron and its distribution within the cell and the TME (Figure 1).

### 4.1. Implication of Lipocalin-2 in Apoptosis

Devireddy and colleagues identified Lcn-2 as one of the death-promoting genes that is transcriptionally activated in IL-3-deprived hematopoietic cells, specifically the pro-B-lymphocytic cell line FL5.12. However, Lcn-2 did not promote apoptosis in all leukocytic cells, as nonhematopoietic cells and monocyte-derived macrophages were resistant, suggesting that the response is cell-type specific [19]. As the intracellular iron concentration increases upon uptake of Lcn-2 via endocytosis, researchers further investigated whether the iron status of the ligand affects the cell fate, or whether cell specificity plays a bigger role [50]. Based on this work, a model was proposed by Richardson, whereby the fate of the cells depends on the Lcn-2-iron status. When Lcn-2 is bound to the iron–siderophore complex, also known as holo-Lcn-2, the cell will internalize the ligand and the receptor, leading to a decrease in TfR and an increase in FT levels, respectively. Donated iron will prevent apoptosis by lowering the expression of the pro-apoptotic protein Bim. On the other hand, apo-Lcn-2 can also be internalized, but it will take a different route compared to the holo-Lcn-2, whereby it will bind inside the cell to the iron–siderophore complex and will subsequently be released via exocytosis. This is suggested to lead to the upregulation of Bim, and to apoptosis of the cell [123].

Rahimi et al. showed that knocking out Lcn-2 in the PC3 prostate cancer cell line, using CRISPR/Cas9, not only decreased PC3 cell proliferation and increased its sensitivity to cisplatin, but also resulted in an enhancement of cisplatin-induced apoptosis [124]. Further, a study by Han and colleagues showed similar effects in gastric cancer cells (MGC-803), whereby the cancer cells were treated with Lcn-2-siRNA plasmids, resulting in the inhibition of cell proliferation, decreased expression of NF-κB and B-cell lymphoma-2 (Bcl-2), enhanced apoptosis, and the increased expression of Bcl2-associated X (Bax) [125]. Other studies not related to cancer research also pinpoint the importance of Lcn-2 in apoptosis. A proteomics study conducted on livers from Lcn-2-deficient mice identified seven upregulated and seven downregulated apoptosis-associated proteins, such as a 5.4-fold and 2.56-fold upregulation of Bax and Annexin A1, respectively [126]. Hence, this reinforces the notion that Lcn-2 can affect different factors in the apoptosis pathway. Lin and colleagues also show that the endometrial carcinoma cell line, RL95-2, upon Lcn-2 administration was protected from apoptosis and showed enhanced cell migration [127]. Hippocampal astrocytes, i.e., neuronal immune cells, were shown to also express high levels of Lcn-2 upon methamphetamine exposure, causing neuronal apoptosis. Lcn-2 was also found to be involved in the cardiovascular system, whereby it causes translocation of the proapoptotic protein Bax from the cytoplasm to the mitochondrial membrane, resulting in Lcn-2-induced cardiomyocyte apoptosis. In addition, an increase in intracellular iron after Lcn-2 administration was abolished after iron chelator treatment, thereby preventing Lcn-2-induced cardiomyocyte apoptosis [128]. Further, the treatment of primary microglia with recombinant Lcn-2 induced the deramification of cells, which is a closely related apoptosis-prone phenotype [129]. So, it seems that either the absence of Lcn-2 or internalization of iron-free Lcn-2 can lead to cell death. This could serve as a foundation for targeting Lcn-2 to induce cancer cell apoptosis.

### 4.2. Implication of Lipocalin-2 in Ferroptosis

The concept of ferroptosis was coined by Stockwell and colleagues as a type of cell death biochemically, morphologically, and genetically distinct from necrosis, apoptosis, and autophagy [30]. Indeed, ferroptosis is a non-apoptotic iron-dependent form of cell death characterized by the accumulation of ROS, leading to lipid peroxidation. As the concept of ferroptosis was investigated further, it was reported that molecules such as RSL3, inactivating glutathione peroxidase 4 (GPX4), lead to the accumulation of peroxidized membrane phospholipids. The effect of ferroptosis is mainly reflected in iron homeostasis and lipid peroxidation genes.

The concept of ferroptosis has been investigated in different tumor models, such as pancreatic, gastric, colorectal, breast, lung, ovarian, hepatocellular, and adrenocortical carcinoma, and melanoma [130,131]. However, ferroptosis is still largely unexplored in the context of cancer, and its connection with other types of cell death especially needs further attention. For example, P53, an important regulator of apoptosis, also regulates ferroptosis, since it can downregulate SLC7A11 via X_c,_ resulting in the inhibition of ferroptosis [132]. Moreover, the accumulation of iron in cancer cells limits tumor progression by increasing chemosensitivity due to oxygen radical species formation and ferroptosis. Persister cells, or drug-resistant cancer cells, express increased levels of GPX4, a protein involved in protection against iron-dependent oxidative stress, as well as increased levels of CD44, a receptor involved in hyaluronate-bound iron uptake. This leads to the suggestion that cancer cells are more susceptible to iron-related perturbations mediated by oxidative stress [133,134]. Additionally, immune cells, known to provide cancer cells with iron, can modulate susceptibility to ferroptosis in the TME. For example, it was observed that CD8^+^ tumor-killing T cells activated by anti-PD-L1 treatment drove the ferroptosis of melanoma and ovarian cancer cells through the secretion of IFNγ, which can downregulate the anti-ferroptotic mediators SLC3A2 and SLC7A11 [135].

Since proteins involved in iron homeostasis, such as FPN, TfR1, and DMT1, influence the development and occurrence of ferroptosis, it can be assumed that Lcn-2 could also play a role in the onset of ferroptosis. Indeed, Chaudhary et al. argue that Lcn-2 can inhibit ferroptosis by stimulating the expression of a component of the cysteine glutamate antiporter, xCT and glutathione peroxidase 4 (GPX4), and by decreasing the intracellular iron levels. By using a monoclonal antibody, they inhibited the function of Lcn-2, which, in turn, decreased the chemo-resistance and tumor progression of colon carcinoma in a xenograft mouse model. The same conclusions were drawn from human tumor samples, whereby xCT and Lcn-2 levels exhibited a positive correlation [136]. The reasoning behind targeting Lcn-2 and not GPX4 and xCT, which are important components of ferroptosis-induced cell death, was the fact that the loss of GPX4 and xCT in mice was either lethal or caused defects in spatial working memory, respectively [137,138]. Interestingly, using RNA sequencing in renal tumor cells, it was reported that upon holo-Lcn-2 administration a cluster of genes implicated in the regulation of ferroptosis, such as solute carrier family 7 member 11 coding for the anionic amino acid transporter light chain (SLC7A11), glutamate–cysteine ligase modifier subunit (GCLM), and glutaminase (GLS), was highly upregulated. The antioxidant *Nrf2* pathway was also triggered in this setup, leading to an increase in ROS, although knocking down *Nrf2* did not induce ferroptosis. Of note, iron delivered by the Tf-TfR system did not activate *Nrf2-*related genes, such as ATP-binding cassette subfamily B member 6 (ABCB6) and ferrochelatase (FECH), despite increasing intracellular iron as efficiently as holo-Lcn-2 [20]. Another study on hepatocellular carcinoma investigated the effect of LIFR, leukemia inhibitory factor receptor, in relation to Lcn-2. NF-κB repression, caused by the interaction of LIFR with SHP1, cytosolic tyrosine phosphatase, repressed Lcn-2 expression, which in LIFR-deficient mice caused severe liver tumorigenesis, the downregulation of iron levels, and resistance to ferroptosis. These authors also discovered that sorafenib, an antineoplastic drug, was more potent when administered in combination with anti-Lcn-2 neutralizing antibodies, which lead to an increase in lipid peroxidation and ferroptosis in tumor tissue [139]. Although the number of discovered ferroptosis regulators is increasing, further research is needed to translate this knowledge into clinical benefit. Nevertheless, it seems that targeting Lcn-2, alone or combined with other targets, could be considered a good option to induce the ferroptosis of cancer cells.

## 5. Lipocalin-2 as a Potential Therapeutic Target

Although Lcn-2 is shown to be highly expressed in certain carcinomas, using Lcn-2 as a therapeutic target has only been tried in the early stages of tumor development. Some of the current strategies to target Lcn-2 involve (i) gene editing techniques, (ii) protein regulation by using antibodies and small-molecule inhibitors, (iii) targeting Lcn-2-related pathways, and (iv) post-transcriptional regulation through RNA interference (Figure 2).

The relationship between TNBC and Lcn-2 was studied, whereby a tumor-targeted nanolipogel (tNLG) successfully knocked out Lcn-2 via the CRISPR technique from human TNBC cells, leading to a significant decrease in TNBC aggressiveness via modulation of epithelial-to-mesenchymal transition and migration, which resulted in an overall smaller tumor growth [140].

Anti-Lcn-2 antibodies have been developed in the last two decades, with the main focus on the destabilization of the Lcn-2/MMP9 complex. Leng and colleagues based their study on a spontaneous mouse breast cancer model (using transgenic mice carrying the mutant form of ErbB2(V664E) driven by the mammary-specific promoter MMTV), in which they observed significantly delayed lung metastasis, and lowered MMP9 expression and activity, in Lcn-2-deficient mice. Next, they expanded their study by using polyclonal anti-Lcn-2 antibodies in an aggressive mouse 4T1-induced mammary tumor model and showed strong interreference with lung metastasis, yet almost no effect on the primary tumor growth [141]. This could be due to the size of the antibody that prevents achieving a sufficiently high concentration within the TME. Although iron chelators and anti-MMP9 moieties have continued to be studied separately, there is a need to further evaluate anti-Lcn-2 antibodies in clinical trials, and in combination with current therapies for specific cancer subtypes.

Targeting Lcn-2-related pathways focuses mainly on breast carcinoma studies. In this context, the downregulation of the NFAT3 transcription factor involved in the anti-invasive and anti-migratory phenotype of breast cancer was found to result in a threefold increase in Lcn-2 expression. This increased expression of Lcn-2 further upregulated the invasion and migration capacity of different estrogen receptor α (ERA+) breast cancer cells, such as BT-474, MCF-7, ZR-75-1, and T-47D. Although the addition of recombinant Lcn-2 to ERA+ cells was sufficient to rescue the inhibition of migration elicited by NFAT3, it did not affect NFAT3′s actin reorganization, which could imply a different migration-regulating mechanism targeted by Lcn-2 [142]. Interestingly, one study by Gwira et al. suggests that Lcn-2 can trigger the activation of the ERK pathway, but this needs to be investigated in depth [143]. Another group studied the effect of the inhibition of the Lcn-2-targeted pathway using ER-negative (ER-) breast cancer and revealed that the transcription factor CCAAT enhancer-binding protein ζ (C/EBP ζ) plays a role in Lcn-2 expression. Indeed, the overexpression of C/EBP ζ in MDA-MB-231 cells resulted in decreased MMP9 and Lcn-2 expression, coinciding with the inhibition of the migration and invasion of breast cancer. Also, C/EBP ζ was found to directly repress human Lcn-2 gene promoter activity by inhibiting Lcn-2 transcription [144]. Hence, this provides further evidence that blocking Lcn-2 could be considered a novel strategy for breast cancer therapy.

Furthermore, Lcn-2 was used as a target in a study by Santiago-Sanchez and colleagues, focusing on IBC and modulating Lcn-2 using small interference RNAs (siRNAs) and inhibitors. Although Lcn-2 CRISPR knock-out TNBC cells did not show a difference in cell proliferation, siRNA-mediated Lcn-2 silencing in IBC cells significantly reduced their viability, invasion, proliferation, and migration [145]. Moreover, in both cholangiocarcinoma (CCA) and breast cancer cells, the siRNA technology to silence Lcn-2 was tested, whereby in the former study, the knock-down of Lcn-2 in the cholangiocarcinoma cell line RMCCA-1 reduced its metastatic properties in vitro. In addition, cell invasion, migration, Lcn-2/MMP9 complex expression, and pro-MMP9 activities were found to be significantly decreased in the manipulated cells [150]. Another study by Guo et al. tested Lcn-2 silencing in combination with targeting C-X-C chemokine receptor type 4 (CXCR4) using liposomes in metastatic TNBC models, MDA-MB-436 and MDA-MB-231, and showed decreased migration of the cells in vitro [151].

## 6. Conclusions and Future Perspectives

Numerous studies highlight that increased Lcn-2 expression could be considered as a very important sensor of cell damage, inflammation, and stress. The fact that it is, on one hand, quickly secreted by innate immune cells upon bacterial infection, while on the other hand, plays a detrimental role in the later stages of tumor development, indicates that Lcn-2 is a very crucial molecule in the immune system. Though several studies discovered a beneficial effect of Lcn-2 during tumor progression depending on the type of tumor, even more studies revealed that Lcn-2 has a rather detrimental effect within the TME and in metastasis.

Hence, further work is needed to broaden our understanding of the exact involvement and mechanisms of action of Lcn-2 in disease progression. More specifically, how exactly macrophages, especially TAMs, are affected by Lcn-2 and changes in Lcn-2-mediated iron deregulation needs further investigation. Secondly, it is of great interest to expand our knowledge on the crosstalk, mediated via Lcn-2, between cancer cells and their environment in the context of iron regulation, as this could be used to specifically target the cancer cell internalization of Lcn-2 and thus block iron take-over by tumor cells. Finally, the fact that Lcn-2 is correlated with both apoptosis and ferroptosis should be utilized as a potential novel targeting strategy to block cancer progression, especially for individuals who exhibit a limited response to immunotherapy. Therefore, we believe that Lcn-2 could be considered as a potentially important targeting molecule, either as a monotherapy or in combination with already existing therapies, whereby novel Lcn-2-targeting approaches might benefit patients with the worst prognosis.

## Figures and Tables

**Figure 1 cancers-15-05159-f001:**
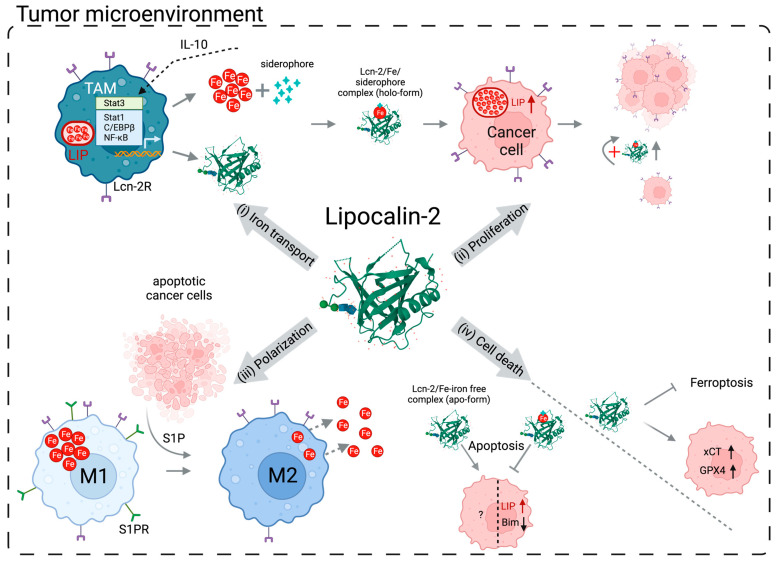
Overview of Lcn-2’s mode of action and effect on (i) iron transport, (ii) cancer cell proliferation, (iii) macrophage polarization, and (iv) cell death in the TME. (i) Lcn-2 expression induced either via the IL-10/STAT3 axis or via different pro-inflammatory cytokines, and iron export, are upregulated by M2-like TAMs, thereby decreasing the LIP. (ii) Siderophore-bound iron is transported and internalized by cancer cells via Lcn-2 (i.e., holo-Lcn-2) and Lcn-2R (24p3R), respectively. Increased LIP inside cancer cells, among other factors, enables cells to proliferate, whilst holo-Lcn-2 continuously provides iron. (iii) M1- and M2-like macrophages, both present in the TME, throughout the course of tumorigenesis are under the influence of different factors, such as S1P, released by apoptotic cancer cells. Upon the binding of S1P to S1PR, macrophages will acquire an iron-releasing anti-inflammatory phenotype. (iv) Finally, Lcn-2 affects apoptosis and ferroptosis by obtaining different formats, whereby the internalization of apo- versus holo-Lcn-2 regulates cell iron availability. Abbreviations: Lcn-2—Lipocalin-2, TME—tumor microenvironment, IL-10—interleukin 10, STAT3—signal transducer and activator of transcription 3, TAM—tumor-associated macrophage, LIP—labile iron pool, Lcn-2R (24p3R)—Lipocalin-2 receptor, holo-Lcn-2—Lcn-2 bound to iron–siderophore complex, S1P—sphingosine-1-phosphate, S1PR—sphingosine-1-phosphate receptor, and apo-Lcn-2—Lcn-2 not bound to iron–siderophore complex. Image created using BioRender.com.

**Figure 2 cancers-15-05159-f002:**
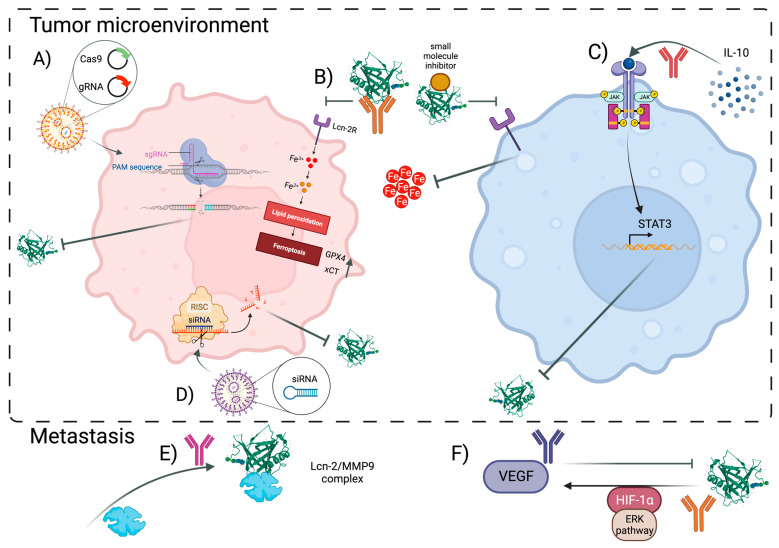
Overview of the potential targeting strategies to block Lcn-2 functionality in both cancer cells (**left panel**) and macrophages (**right panel**) in the TME, but also to prevent metastasis. (**A**) CRISPR/Cas9 editing technique together with optimized nanocarriers can be used in research to study cancer cell-derived effect of Lcn-2 in tumor progression, and in the future as a therapeutic strategy. (**B**) Lcn-2 can be blocked by monoclonal antibodies and small-molecule inhibitors such as ZINC00784494 or ZINC00640089, potentially affecting iron availability, whereby the induction of ferroptosis in cancer cells, and stronger iron sequestration in macrophages, could be triggered. (**C**) Due to the strong upregulation of Lcn-2 upon IL-10 (an anti-inflammatory cytokine highly abundant in the TME) stimulation by macrophages, via STAT3 and C/EBPβ, it could be suggested that blocking IL-10 could reduce the expression of Lcn-2, ultimately preventing macrophage polarization towards an M2-like phenotype. (**D**) Similarly, as with (**A**), siRNA-mediated Lcn-2 could help unravel the role of Lcn-2 in carcinogenesis, which could be further translated into combinatorial therapy. Finally, since Lcn-2 contributes to invasion and metastasis, and angiogenesis, by binding to MMP9 and inducing HIF-1α and VEGF, respectively, targeting these complexes could be utilized to prevent further tumor development. (**E**) Destabilization of Lcn-2/MMP9 complex carcinoma led to decreased lung metastasis, while (**F**) blocking VEGF affected angiogenic activity of Lcn-2. However, Lcn-2 also induces VEGF mediated by HIF1α via the Erk pathway, which could be another Lcn-2-dependent mechanism used as a therapeutic approach [141,145,146,147,148,149]. Abbreviations: Lcn-2: Lipocalin-2, TME—tumor microenvironment, CRISPR/Cas9—clustered regularly interspaced palindromic repeats/CRISPR-associated protein 9, STAT3—signal transducer and activator of transcription 3, C/EBPβ—CCAAT enhancer-binding proteins β, MMP9—matrix metalloprotease 9, HIF1α—hypoxia-inducible factor 1α, and VEGF—vascular endothelial growth factor. Created using BioRender.com.

## Data Availability

The data presented in this study are available in this article.

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
