# Peer review of "Lipocalin-2: A Nurturer of Tumor Progression and a Novel Candidate for Targeted Cancer Therapy"

_cancers, 2023, doi:10.3390/cancers15215159_

Round 1

Reviewer 1 Report

In the current manuscript ‘Lipocalin-2: a nurturer of tumor progression and a novel candidate for targeted cancer therapy’ by Živalj et al. a very detailed and comprehensive overview about the role of lipocalin-2 during tumor progression is given. Containing also therapeutic approaches and future research that is needed. The review is well structured and provides new information together with a summary what is known about lipocalin-2 and its role in the tumor microenvironment.

Only a few minor points could be considered by the authors:

1.   In line 70 to 72 the authors describe lipocalin-2 as an alternative regulator of iron-homeostasis playing a role in ferroptosis. It might me helpful for the reader to work out more the link between iron and ferroptosis already at this point. In addition, a more detailed statement/hypothesis for the role lipocalin-2 or its iron-transporting function could play in this setup might also be helpful to underline this novel and important function also giving a glimpse for the upcoming chapter in this review.   

2.   In line 102-103 the authors talk about the association of anemia and cancer. Adding one or two sentences about anemia of inflammation might also be useful to link the inflammatory properties of lipocalin-2 usually occurring during infections with its role in the tumor microenvironment. Later the authors also referred to this aspect, for example in line 141, 293.

Only minor english editing is needed. 

Author Response

Reviewer 1

In the current manuscript ‘Lipocalin-2: a nurturer of tumor progression and a novel candidate for targeted cancer therapy’ by Živalj et al. a very detailed and comprehensive overview about the role of lipocalin-2 during tumor progression is given. Containing also therapeutic approaches and future research that is needed. The review is well structured and provides new information together with a summary what is known about lipocalin-2 and its role in the tumor microenvironment.

We thank the reviewer for the very supporting comment.

Only a few minor points could be considered by the authors:

  1. In line 70 to 72 the authors describe lipocalin-2 as an alternative regulator of iron-homeostasis playing a role in ferroptosis. It might be helpful for the reader to work out more the link between iron and ferroptosis already at this point. In addition, a more detailed statement/hypothesis for the role lipocalin-2 or its iron-transporting function could play in this setup might also be helpful to underline this novel and important function also giving a glimpse for the upcoming chapter in this review.

This is a very good suggestion of the reviewer. We have included a few sentences to include the connection between iron, ferroptosis and Lcn-2 (lines 74-80). In addition, the reference list has been updated (see lines 532-536).

  1. In line 102-103 the authors talk about the association of anemia and cancer. Adding one or two sentences about anemia of inflammation might also be useful to link the inflammatory properties of lipocalin-2 usually occurring during infections with its role in the tumor microenvironment. Later the authors also referred to this aspect, for example in line 141, 293.

In the revised version of we have elaborated on the concept anemia of inflammation and brought this in the context of cancer development and how Lcn-2 can further fuel the development of anemia in cancer patients.

Main adaptations are:

Lines 115-121: “…, cancer-related anemia (CRA) is a cytokine-mediated disorder resulting from complex interactions between tumor cells and the immune system and is characterized by biological and hematologic features that resemble those described in anemia associated to chronic inflammatory disease (i.e. anemia of inflammation). It was also shown that the release of certain inflammatory cytokines during anemia of inflammation negatively affects erythroid progenitor cell differentiation.

Lines 123-126: “…, Lcn-2 through its iron regulating role could be implicated as a pivotal player in CRA by regulating the inflammatory immune response within the TME and iron availability for erythropoiesis.

Additionally, the reference list has been updated whereby the references have been added in lines: 857-867, 872-875.

Only minor English editing is needed.

Throughout the manuscript, we have improved the quality of the written English.

Reviewer 2 Report

This review is devoted to summarizing the current data on the role of lipocalin-2 in tumor development, including the protein's role in regulating iron availability to tumor cells, its effect on iron transfer from macrophages and their polarization of macrophages, modulation of programmed cell death, in particular ferroptosis, and finally ways to target lipocalin-2 as a therapeutic target. The authors operate with contemporary literature, although it is not entirely clear what criteria were used to select the literature, whether keyword sifting was performed, or whether other criteria for inclusion or exclusion of literature sources were used. However, this observation does not diminish the appreciation of the quality of the review and the thoroughness of the authors' consideration of the problem. The style of presentation and logic appear scholarly and make a good impression. We would like to emphasize the good quality and clarity of the illustrations. In my opinion, the review can be published in the form presented.

Author Response

Reviewer 2

This review is devoted to summarizing the current data on the role of lipocalin-2 in tumor development, including the protein's role in regulating iron availability to tumor cells, its effect on iron transfer from macrophages and their polarization of macrophages, modulation of programmed cell death, in particular ferroptosis, and finally ways to target lipocalin-2 as a therapeutic target. The authors operate with contemporary literature, although it is not entirely clear what criteria were used to select the literature, whether keyword sifting was performed, or whether other criteria for inclusion or exclusion of literature sources were used. However, this observation does not diminish the appreciation of the quality of the review and the thoroughness of the authors' consideration of the problem. The style of presentation and logic appear scholarly and make a good impression. We would like to emphasize the good quality and clarity of the illustrations. In my opinion, the review can be published in the form presented.

We thank the reviewer for the very supporting comment.